# Biomechanical Loading Comparison between Titanium and Bioactive Resorbable Screw Systems for Fixation of Intracapsular Condylar Head Fractures

**DOI:** 10.3390/ma13143153

**Published:** 2020-07-15

**Authors:** Shintaro Sukegawa, Norio Yamamoto, Keisuke Nakano, Kiyofumi Takabatake, Hotaka Kawai, Takahiro Kanno, Hitoshi Nagatsuka, Yoshihiko Furuki

**Affiliations:** 1Department of Oral and Maxillofacial Surgery, Kagawa Prefectural Central Hospital, 1-2-1, Asahi-machi, Takamatsu, Kagawa 760-8557, Japan; furukiy@ma.pikara.ne.jp; 2Department of Orthopaedic Surgery, Kagawa Prefectural Central Hospital, Takamatsu, Kagawa 60-0065, Japan; norio-yamamoto@umin.ac.jp; 3Department of Oral Pathology and Medicine, Okayama University Graduate School of Medicine, Dentistry and Pharmaceutical Sciences, Okayama 700-8530, Japan; pir19btp@okayama-u.ac.jp (K.N.); gmd422094@s.okayama-u.ac.jp (K.T.); de18018@s.okayama-u.ac.jp (H.K.); jin@okayama-u.ac.jp (H.N.); 4Department of Oral and Maxillofacial Surgery, Shimane University Faculty of Medicine, Shimane 693-0021, Japan; tkanno@med.shimane-u.ac.jp

**Keywords:** titanium screw, bioactive resorbable screw, biomechanical loading evaluation, fracture fixation, intracapsular condylar head fracture

## Abstract

Osteosynthesis resorbable materials made of uncalcined and unsintered hydroxyapatite (u-HA) particles, poly-L-lactide (PLLA), are bioresorbable, and these materials have feasible bioactive/osteoconductive capacities. However, their strength and stability for fixation in mandibular condylar head fractures remain unclear. This in vitro study aimed to assess the biomechanical strength of u-HA/PLLA screws after the internal fixation of condylar head fractures. To evaluate their biomechanical behavior, 32 hemimandible replicas were divided into eight groups, each consisting of single-screw and double-screw fixations with titanium or u-HA/PLLA screws. A linear load was applied as vertical and horizontal load to each group to simulate the muscular forces in condylar head fractures. Samples were examined for 0.5, 1, 2, and 3-mm displacement loads. Two screws were needed for stable fixation of the mandibular condylar head fracture during biomechanical evaluation. After screw fixation for condylar head fractures, the titanium screws model was slightly more resistant to vertical and horizontal movement with a load for a small displacement than the u-HA/PLLA screws model. There was no statistically significant difference with load for large displacements. The u-HA/PLLA screw has a low mechanical resistance under small displacement loading compared with titanium within the limits of the mandibular head fracture model study.

## 1. Introduction

Mandibular condylar fractures are one of the most common types of maxillofacial fractures, and mandibular head fractures are a frequent disease that accounts for about half of all mandibular condylar fractures [1]. Neff et al. [2] classified mandibular condylar head fractures into three types: type A (a fracture line through the medial portion of the condylar head), type B (a fracture line through the lateral portion of the condylar head), and type C (a fracture line is near the attachment of the lateral capsule). Mandibular condylar head fractures are alternatively treated either by closed treatment or open reduction and internal fixation (ORIF) [3]. Surgical treatment recommended ORIF for types B and C condylar fractures with the shortening of mandibular ramus height [4,5]. Various forms of titanium osteosynthesis for the internal fixation of condylar head fractures have been applied such as microplates, miniplates, screws, and small-fragment positional screws [4,5,6,7].

Among a large number of available types of fixators for the surgical treatment of condylar head fractures, screw fixation is the most widely used technique. It is minimally invasive, anatomically based, and has a number of advantages compared with the use of a plate system [8,9]. Surgical treatment with titanium screws is very helpful. However, this osteosynthesis remains in the human body and may need to be removed later due to complications [7]. Secondary surgery for removal is stressful for patients and surgeons, so it is desirable to avoid this as much as possible.

Since resorbable osteosynthesis material is degraded in vivo [10], secondary surgery for removal is basically unnecessary. Many resorbable osteosynthetic materials are currently commercially available. In recent years, osteosynthesis materials having bioactivity as well as absorption characteristics have been developed and used in clinical practice. In particular, bioactive/osteoconductive and bioresorbable osteosynthesis devices made from a composite of uncalcined and unsintered hydroxyapatite (u-HA) particles and poly-L-lactide (PLLA) are excellent [11]. This device has biocompatible and osteoconductive characteristics and is very advantageous as an osteosynthesis material [12,13]. However, both the strength and stability of the osteosynthesis material for fixation to the mandibular condylar head fracture are still unclear. Thus, the aim of this in vitro study was to assess the biomechanical strength of u-HA/PLLA bioactive, resorbable osteosynthesis systems after the fixation of mandibular condylar head fracture.

## 2. Materials and Methods

### 2.1. Materials

In this study, we used the bioactive resorbable osteosynthesis systems, Super FIXSORB MX^®^ (Teijin Medical Technologies Co., Ltd., Osaka, Japan). Forged composites of u-HA/PLLA were processed by machining or milling treatments to form screws, which contained 30% weight fractions of u-HA particles. u-HA is raw HA that contains neither calcined nor sintered material. Composite screws were filled with 30% weight fractions of u-HA particles. The screws used in this study were 2.0 mm in diameter. A 2.0-mm screw system (MatrixMANDIBLE™; DePuy Synthes, Raynham, MA, USA) was used as the titanium screw. These titanium screws are alloys and titanium grade 4.

### 2.2. Evaluation of Shear Strength of Titanium and Bioresorbable Screw

#### Sample Preparation and Measurement

The shear test jig in this study contained three parts. In the shear test, a hole of 2.5 mm diameter was used, and three parts were combined so that the holes cooperate with each other. The screw was first inserted to measure into the hole. Since the screws are sheared on both sides of the loading jig, the total test force at the two locations was measured. Therefore, the shear strength of the screw was half the test force. Using the method illustrated in Figure 1, the measured screws were as follows: (1) u-HA/PLLA bioresorbable screw (diameter, 2 mm; length, 12 mm); (2) titanium screws (diameter, 2 mm; length, 12 mm). The shear strength test was performed five times each. The maximum stress and the displacement at the maximum stress were recorded.

### 2.3. Biomechanical Loading Evaluation

#### 2.3.1. Sample Preparation

We used 32 polyurethane replicas of human hemimandibles (Mandible, Code #8900; SYNBONE AG, Laudquart, Switzerland). Although a polyurethane mandible replicates the property of cancellous bone [14], this model is useful to obtain preliminary results on the stability of the osteosynthesis system for fractures of the condylar head, which is dominated by cancellous bone [15]. For modeling mandibular condylar head fractures, we used the Neff classification: the most recognized and adopted by the Strasbourg Osteosynthesis Research Group [2]. Type B fracture patterns usually lead to shortening and malocclusion of the associated mandibular ramus, which usually requires ORIF [3]. For an ordinary mandibular condylar head fracture type B model, we used a fracture line that starts slightly outside the attachment of the lateral capsule and continues outside the condyle and created a fracture in the left condyle of each model. These mandibular condylar head fracture models were created using a computer-controlled program with cutting guides. A partial cut was made in each hemimandibular model using diamond disks (KG Sorensen, Cotia, São Paulo, Brazil). A silicone mold of the condylar head of the osteosynthesis was created in the mandible and guide holes were created with a drill; then, using the mold as a guide, we completed the cut. This method was used to create identical cuts and perforations in all the left hemimandibular condylar head fracture models, thus guiding the positions for screw fixation for each study group (Figure 2A).

#### 2.3.2. Fixation Method for Mandibular Condylar Fracture

Osteotomized mandibles were randomly divided into the following four groups and used for different fixation systems: (1) single titanium screw, (2) single u-HA/PLLA bioresorbable screw, (3) double titanium screws, and (4) double u-HA/PLLA bioresorbable screws.

The screws used were titanium (2.0 mm diameter × 12 mm long) or u-HA/PLLA (2.0 mm diameter × 12 mm long). The fracture line and screw insertion direction were set according to the report of Guo et al. [5]. The points were fixed as follows: (1) the upper end of the mandible condylar fracture line, (2) the medial inferior point of the fracture surface, and (3) the innermost point of the sagittal fragment.

The direction of passing through the midpoint of points 1 and 2 toward 3 was the direction of screw insertion. Double-screw fixation was moved parallel to the caudal side from this direction, and the other was inserted (Figure 2B).

#### 2.3.3. Biomechanical Loading Test

After screw fixation of the fracture segment, replicas were mounted on a testing machine (AG-20KNX; Shimazu, Kyoto, Japan) based on a biomechanical cantilever-bending model that simulates masticatory forces. Then, the mandibular ramus and angle areas of each replica were stabilized. The adaptation of the semi-fixed polyurethane to the machine was guaranteed by the metal support. Based on past biomechanical evaluation methods [16], the replicas were then subjected to linear loading in two directions: from lateral to medial (horizontally; Figure 3A) and from anterior to posterior (vertically; Figure 3B). These forces simulated functional masticatory forces applied to an actual fractured condylar head. Thus, this material testing unit created linear displacement at a rate of 1 mm/min, and loading continued until the maximum load was reached. The peak load and displacement for each fracture model were recorded. All fracture models were analyzed for displacements of 0.5, 1.0, 2.0, and 3.0 mm by loading and for the amount of displacement by the maximum load. Means and standard deviations were derived from each test and evaluated for statistical significance.

### 2.4. Statistical Analysis

Data were recorded and entered into an electronic database throughout the study using Microsoft Excel (Microsoft Inc., Redmond, WA, USA). Means and standard deviations (SD) were used where distribution was compatible with normality. The database was transferred to JMP version 14.2.0 software for Macintosh computers (SAS Institute Inc., Cary, NC, USA) for statistical analysis. Student’s *t*-tests for independent samples were performed to investigate whether the significance of differences between the mean values of the each groups. A *p*-value of <0.05 was considered statistically significant.

## 3. Results

### 3.1. Strength Evaluation

The shear test force was 1156.0 N (SD, ±386 N) for the titanium screw and 184.6 N (±18.7 N) for the u-HA/PLLA screw (Figure 4). In the shear test, the titanium screw was significantly stronger than the u-HA/PLLA bioresorbable screw (*p* < 0.05). The u-HA/PLLA bioresorbable screw shear resistance was only 16.0% that of the titanium screw.

### 3.2. Biomechanical Loading Evaluation

#### 3.2.1. Horizontal Loading Test

Compared with u-HA/PLLA screws, both double-fixed screws showed statistically significantly higher strength at large displacements (Figure 5). In all cases, rotation of the fractured piece occurred near the maximum force used in the test. On the other hand, double-fixed screws did not cause the rotation of the fractured piece (Figure 6). The measured values were most prominent at 3-mm displacement: u-HA/PLLA double-fixed screws had 1.25 times higher resistance than single-fixed screws, whereas titanium double-fixed screws had 1.32 times higher resistance than single-fixed screws.

A comparison of titanium and u-HA/PLLA screws with two screws in a horizontal load test is shown in Figure 7. Similar to the vertical load test, the resistance value of the titanium screw was higher than that of the u-HA/PLLA screw at 0.5-mm and 1-mm displacements. The maximum resistance values were 0.5 and 1 mm, which were 1.24 and 1.17 times that of u-HA/PLLA, respectively. However, there was no significant difference between the two screws at 2-mm and 3-mm displacements.

#### 3.2.2. Vertical Loading Test

The results of this experiment demonstrated that the mechanical resistances among osteosynthesis screws remained proportional to the amount of displacement. When comparing the single and double titanium screws and the u-HA/PLLA screw, the double-fixed screws showed significantly higher strength in titanium than the u-HA/PLLA screws (Figure 8). The measured values were most prominent during the initial displacement; the u-HA/PLLA double-fixed screw had 1.36 times higher resistance than the single-fixed screw and the titanium double-fixed screw had 1.30 times higher resistance than the single-fixed screw at 0.5-mm displacement. Figure 9 shows a comparison between titanium and u-HA/PLLA screws with double fixation. The resistance value of the titanium screw was significantly higher than that of the u-HA/PLLA screw at displacements of 0.5 mm and 1 mm. However, there was no significant difference between the two screws at displacements of 2 mm and 3 mm.

## 4. Discussion

This study is the first report of an in vitro biomechanical evaluation using bioresorbable screws for the osteosynthesis of mandibular condylar head fractures. In this study, the bioactive u-HA/PLLA screw was lower in strength than conventional titanium screws. On the other hand, in a comparison after two-screw fixations for mandibular condylar head fractures, the bioresorbable screw model had slightly lower resistance to the vertical and horizontal loads for initial displacement than that for the titanium screw model in biomechanical evaluation. By contrast, there was no statistically significant difference in the load for large displacement. These results suggest that bioactive u-HA/PLLA screws have a slightly lower mechanical resistance compared with the titanium screws within the limitations of a mandibular condylar head model study.

Screw osteosynthesis for mandibular joint fractures is a relatively minimally invasive operation and is the current standard surgical treatment for this type of fracture. The primary reason for this is that screw fixation provides better fixation stability, a smaller incision area, shorter surgery time, and less scarring of the temporomandibular joint than plate fixation [17]. The second reason is that screws can be applied without interfering with soft tissue. Fractures of the mandibular head always move anteriorly, inferiorly, and medially because of the contraction of the lateral pterygoid muscle. This dislocation causes shortening of the mandibular ramus and bad occlusion. However, the lateral pterygoid muscle must be preserved because it plays an important role in fracture healing [18,19]. Screw fixation, which has a small working space and can reduce and fix a displaced bone fragment, is also useful for healing. However, a titanium screw that has completed its role after osteosynthesis remains indefinitely inside the human body. Even if there is no problem immediately after the operation, a screw that has remained in the body may later cause complications such as those associated with the stress-shielding effect, condylar head resorption, perforation of the articular fossa, or exposure of the pin during remodeling of the bone [20,21]. Therefore, fixation using resorbable screws is desirable for mandibular condylar head fracture. In addition, it would be ideal if the absorbent material used had bioactivity.

Bioabsorbable materials are used in maxillofacial surgery, including orthognathic surgery [22], the osteosynthesis of maxillofacial fractures [23], orbital reconstruction [24,25], bone augmentation of dental implants [12], reconstruction after maxillofacial cysts, and tumor removal [26,27]. Although the clinical usefulness of bioresorbable materials has been reported, its insufficient strength is a problem for use in sites where a functional load is applied, such as the jaw. Unlike titanium, bioresorbable materials generally cannot withstand high mechanical loads [16]. In this study, we first examined the mechanical strength of titanium and u-HA/PLLA bioresorbable screws. The strength of the u-HA/PLLA bioresorbable system for titanium screws was about 60% in torsional strength and surprisingly only about 16% for shear strength. These results were similar to strength research using bioresorbable screws of the same composition [28]. Thus, in regard to pure biomechanical strength, our findings may suggest that u-HA/PLLA bioresorbable screws are much weaker than titanium screws.

In our biomechanical test study, we examined the number of screws for fixation with one and two screws, respectively. Titanium double fixation showed a maximum load value that was about 1.3 times higher than the single fixation of their strength in vertical load and horizontal load tests. The u-HA/PLLA double fixation showed about 1.2–1.4 times higher resistance than single fixation in vertical load and horizontal load tests. This suggests that the double fixation increased the fixing force against loads in the both vertical and horizontal directions. As a noteworthy point, fixation with a single screw caused the bone fragments to turn under a high load in the horizontal load test. This is consistent with previous reports suggesting that using a single screw for fixation is not sufficient to provide rotational stability [29]. The clinical study further considered that a two-screw fixation was stable for the fixation of head fractures due to the superior retention of the cancellous bone in the mandibular condylar head [29]. The results of our study support these previous findings.

Vertical load direction imitates the functional load at the time of mouth opening [30]. The fixation of double u-HA/PLLA screws showed a significantly lower resistance at the initial load compared with the two titanium screws, but this was not the case at higher loads. Therefore, the bioresorbable screw was clinically useful for osteosynthesis in the open treatment of mandibular condylar fractures, but it was considered appropriate to limit the load applied during the initial function. By contrast, the load in the horizontal direction was slightly inferior to that of the titanium screw. This result is very interesting. Although the shear strength of u-HA/PLLA was clearly low, as shown in the first experiment, using double-screw fixation generated high strength. This load direction imitates functional load during the lateral movement of the mandible [16,30]. This result suggested that the bioresorbable screw had sufficient strength for a functional load during the lateral exercise for the osteosynthesis of mandibular condylar head fracture.

Shear strength was presumed to be important for osteosynthesis devices in the mandibular condylar head fractures in this study. Therefore, the titanium screw showed higher strength than the bioresorbable screw even in the biomechanical test due to the excellent shear strength of the titanium screw itself. It is desirable to improve the shear strength to actively use the biomechanical screw in the future.

In this study, biomechanical strength was assessed by measuring the load as a function of the amount of displacement. This study method was based on our previous experimental design in biomechanical studies of mandibular condylar fractures [16]. Small displacements are assumed to occur at rest or at small functions and are involved in the better stability of bone healing. On the other hand, large displacements are assumed to occur during functions such as large mouth opening. In small displacements, the bioresorbable screw fixation showed slightly lower strength than the titanium screw fixation. From our experimental results, maxillofacial surgeons have to devise low strength for clinical application bioresorbable screws. It will be necessary to continue to limit the opening due to traction with rubber. There was no significant difference between bioresorbable and titanium screws in large displacements, but single-screw fixation caused rotational deformation in the horizontal test. Therefore, when selecting a single screw due to anatomical restrictions, strong restrictions such as intermaxillary fixation might be necessary.

It of great importance to determine the strength of the u-HA/PLLA materials used in this study. This is because the most common conventional resorbable material, pure PLLA, has disadvantages such as inability to bind directly to bone, unstable resorption and degradation processes, and long replacement times [31,32]. By contrast, u-HA/PLLA material has high biological activity and advantages such as osteoconductivity, biocompatibility, direct binding to bone, and stable complete degradation and absorption [11,12,13,33,34]. This is the effect of the unfired hydroxyapatite compound. However, synthetic materials can cause a reduction in strength. Thus, this study—in which we clarified its strength as an osteosynthesis material for mandibular condylar head fractures—is significant.

This study has some limitations. As an in vitro biomechanical stress test, it does not fully reflect the dynamics of humans. However, the major functional load of mandibular movement has been reproduced experimentally. This is the first study of osteosynthesis for mandibular condylar head fracture using bioactive u-HA/PLLA screw, and it is important. Second, the mechanical model used in this study, the polyurethane model, is softer than the actual human mandible. The mandibular condylar head is a site with abundant bone marrow and close strength, but it is not enough to reproduce a faithful cortical bone in this model. Therefore, the results of this study suggest that the bioresorbable materials should not be used immediately for clinical treatment. Given the lower biomechanical strength of the u-HA/PLLA screw using this mandibular model, the actual clinical use of the resorbable screw should avoid early overload immediately after surgery. With the use of these u-HA/PLLA screws, patients with strong occlusal forces may need to be restricted by dietary guidance and elastic rubber to eat soft food after surgery.

## 5. Conclusions

In this study, we show that a bioactive bioresorbable u-HA/PLLA screw has lower strength than the conventional titanium screw. Two screws were needed for stable fixation of the mandibular condylar head fracture. However, in a comparison after screw fixation for mandibular condylar head fractures, the titanium screw model was slightly more resistant to vertical and horizontal loads with a load for a small displacement than the u-HA/PLLA screws model in biomechanical evaluation. By contrast, there was no statistically significant difference in load for large displacements. These results suggest that the bioactive bioresorbable u-HA/PLLA screw has low mechanical resistance under small displacement loading compared with the titanium screw within the limits of the mandibular condylar head fracture model.

## Figures and Tables

**Figure 1 materials-13-03153-f001:**
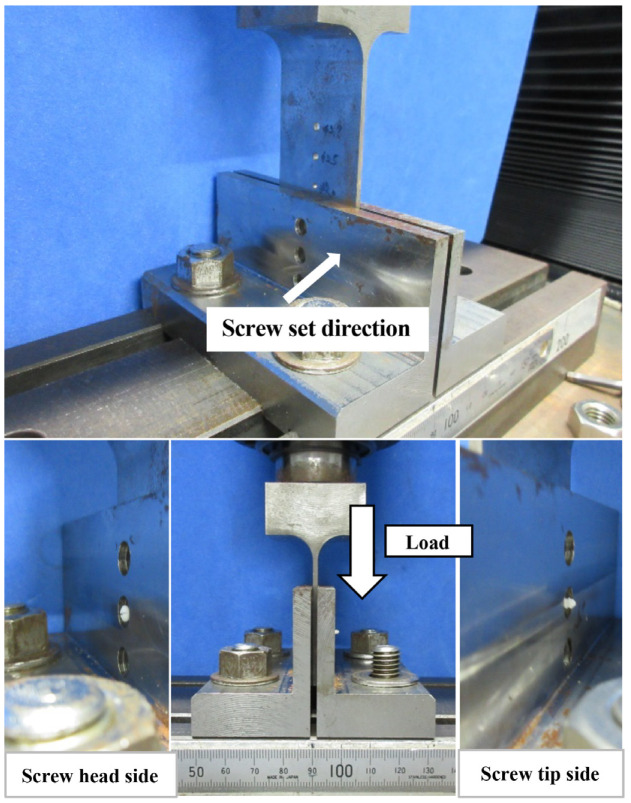
Machine set for measuring screw shear strength. The center hole of the measuring machine has a diameter of 2.5 mm.

**Figure 2 materials-13-03153-f002:**
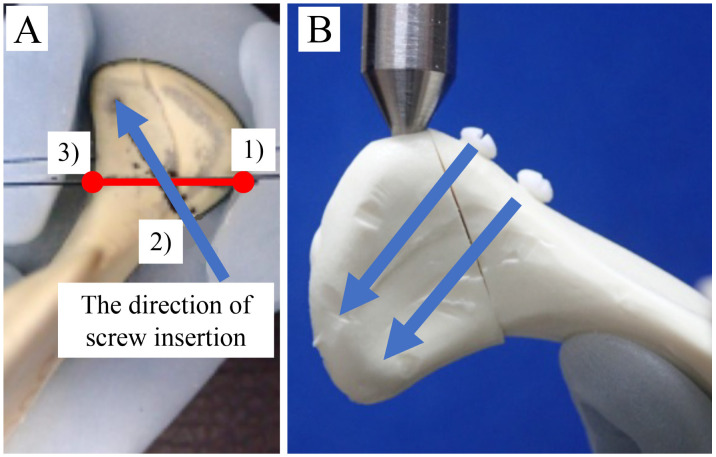
(**A**) A mandibular condylar head fracture type B model, using a fracture line that starts slightly outside the attachment of the lateral capsule and continues outside the condyle, was created as shown in Figure 1. 1) The upper end of the mandible condylar fracture line, 2) The medial inferior point of the fracture surface, and 3) The innermost point of the sagittal fragment. The direction of passing through the midpoint of points 1 and 2 toward 3 was the direction of screw insertion. (**B**) Double-screw fixation was moved parallel to the caudal side from this direction, and the other was inserted.

**Figure 3 materials-13-03153-f003:**
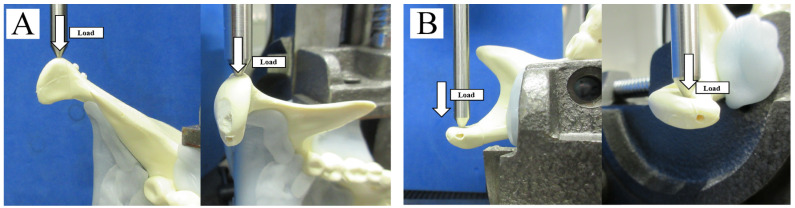
Biomechanical loading test. (**A**) Horizontal (lateromedial) linear loading. (**B**) Vertical (anteroposterior) linear loading.

**Figure 4 materials-13-03153-f004:**
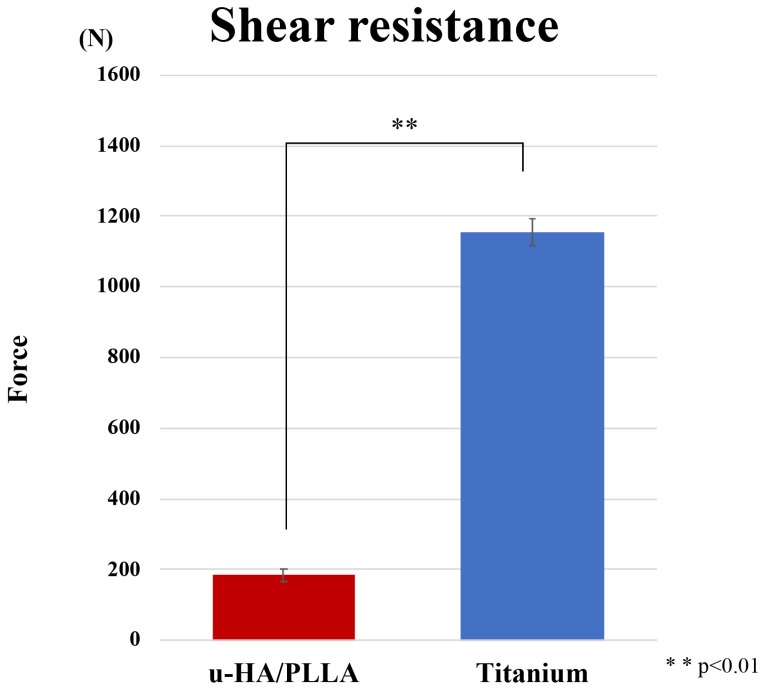
Comparison of titanium screw and unsintered hydroxyapatite (u-HA)/poly-L-lactide (PLLA) bioresorbable screw in the shear test.

**Figure 5 materials-13-03153-f005:**
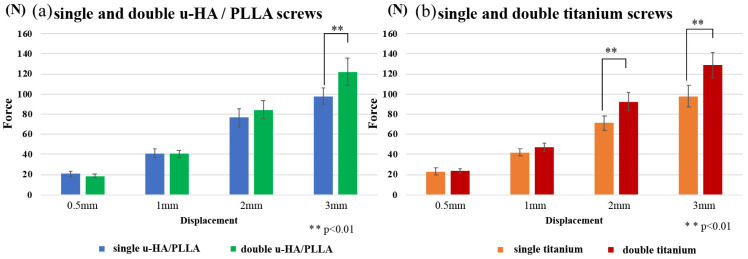
Horizontal loading tests. These graphs show the comparison of single and double u-HA/PLLA screws (**a**) and comparison of single and double titanium screws (**b**).

**Figure 6 materials-13-03153-f006:**
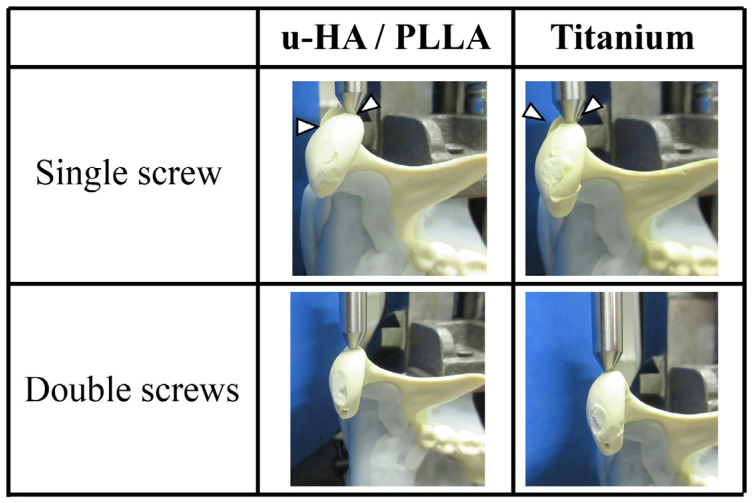
In single-fixed cases, rotation of the fractured piece (arrow head) occurred near the maximum force used in the test. Double-fixed screws did not cause any rotation of the fractured piece.

**Figure 7 materials-13-03153-f007:**
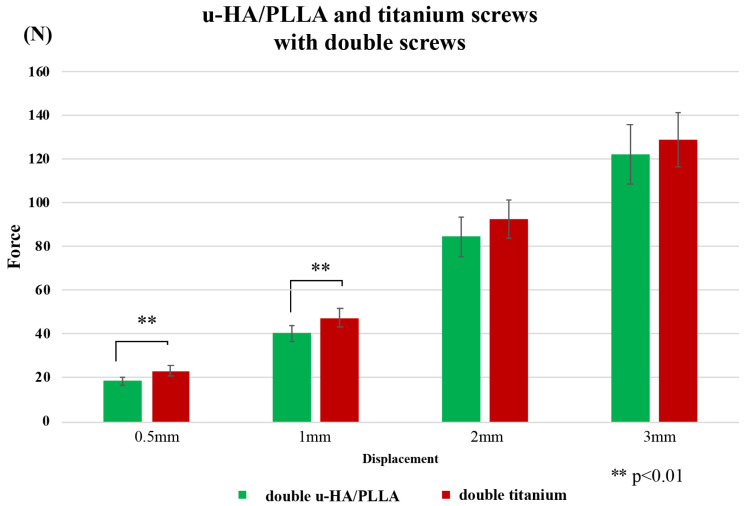
Comparison between titanium and u-HA/PLLA screws with double fixation in the horizontal test.

**Figure 8 materials-13-03153-f008:**
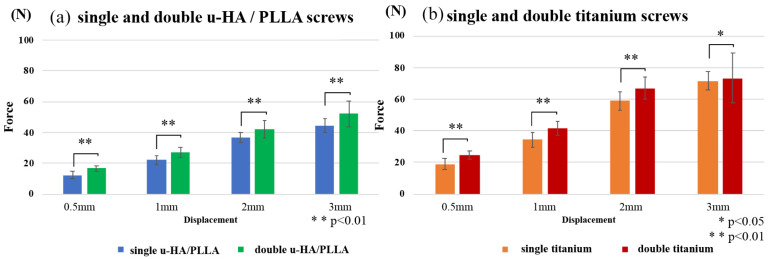
Vertical loading tests. These graphs show the comparison of single and double u-HA/PLLA screws (**a**) and comparison of single and double titanium screws (**b**).

**Figure 9 materials-13-03153-f009:**
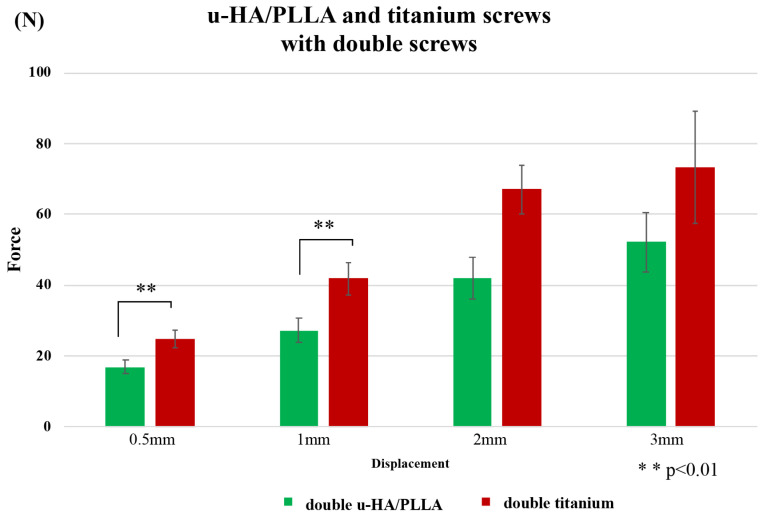
Comparison between titanium and u-HA/PLLA screws with double fixation in the vertical loading tests.

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
