# Peer review of "Biomechanical Loading Comparison between Titanium and Bioactive Resorbable Screw Systems for Fixation of Intracapsular Condylar Head Fractures"

_materials, 2020, doi:10.3390/ma13143153_

Round 1
Reviewer 1 Report
Dear Authors,
Your paper is very intersting to read as I work also on bio-absobable screws.
Nethertheless there could be made some small ameliorations:
- Your fig 2 could be made better by more clear definition of your reference points ( arrows ? ) and mark your lines
- your figure 3 is in my opinion not clearly in accordance with the text - doesn't fig A show the latero-medial loading ? and vice versa ?
- in your fig 6 I miss the mentioned ABCD -at least the A for the rotational deformity
at the end of your discussion session from line 281 you could give your proposals for after treatment - for how long you would like to restrict strong occlusal forces - what would you prefer rubber or only diatary guidance ? how would you contro - by CT 6/12 weeks X-ray only in cases if needed ? - do you see any potential to make the screws stronger
I can not see how or what conclusion you get out of your small or large displacement trials - for me as trauma surgeon I would like to know about more concerning your thougths about this interesting testing results
Author Response
Reviewer 1
Your paper is very intersting to read as I work also on bio-absobable screws.
Nethertheless there could be made some small ameliorations:Nevertheless, there are some aspects that should be clarified or better discussed:
Comment 1) Reviewer1: Your fig 2 could be made better by more clear definition of your reference points ( arrows ? ) and mark your lines
Author response: We thank the reviewer for this helpful comment. As you pointed out, we have defined the reference point more clearly and changed it to a line marked figure.
Comment 2) Reviewer1: Your figure 3 is in my opinion not clearly in accordance with the text - doesn't fig A show the latero-medial loading ? and vice versa ?
Author response: We thank the reviewer for this helpful comment. It is as you pointed out. We have modified the text.
Comment 3) Reviewer1: in your figure 6 I miss the mentioned ABCD -at least the A for the rotational deformity at the end of your discussion session from line 281 you could give your proposals for after treatment - for how long you would like to restrict strong occlusal forces - what would you prefer rubber or only diatary guidance ? how would you contro - by CT 6/12 weeks X-ray only in cases if needed ? - do you see any potential to make the screws stronger I can not see how or what conclusion you get out of your small or large displacement trials - for me as trauma surgeon I would like to know about more concerning your thougths about this interesting testing results
Author response: We thank the reviewer for this helpful comment. We envision full replacement of bioresorbable screws for mandibular head fractures with limited clinical use.
We have added the following to the discussion section:
“In this study, biomechanical strength was measured by measuring the load as a function of the amount of displacement. This study method was based on our previous experimental design in biomechanical studies of mandibular condylar fractures [16]. Small displacements are assumed to occur at rest or at small functions and are involved in better stability of bone healing. On the other hand, large displacements are assumed to occur during functions such as large mouth opening. In small displacements, the bioresorbable screw fixation showed slightly lower strength than the titanium screw fixation. From our experimental results, maxillofacial surgeons have to devise it for clinical application. It will be necessary to continue to limit the opening due to traction with rubber. There was no significant difference between absorbency and titanium in large displacement, but single screw fixation caused rotational deformation in the horizontal test. Therefore, when selecting a single screw due to anatomical restrictions, strong restrictions such as intermaxillary fixation may be necessary. “

Reviewer 2 Report
The paper is a well written and well structured description of a biomechanical comparison of titanium and bioactive resorbable screw systems. A generally thorough literature review demonstrates the premise for the work, it's novelty and importance. It would be beneficial either within the literature review or discussion to mention the work of Pavlychuk et al " A comparative biomechanical evaluation of different osteosynthesis techniques used for intracapsular condylar head fractures" this paper whilst not overriding the novelty of this work does contradict the first statement within the discussion that 'This study is the first report of an in vitro biomechanical evaluation using bioresorbable screws for osteosynthesis of mandibular condylar head fractures.' The detailed comparison of the titanium and bioresorbable screw as well as the comparison of 1 and 2 screws is novel and of significant clinical value.
The method was systematic and well written containing all necessary information except for description in statistics where it should be described how normality of data was determined.
Results were clearly and concisely presented although figure 6 did not adequately show the rotation, a close up of the defect region would be beneficial.
Specific points within the discussion:
Line 205: why does the sentence start with 'However' this sentence as in the previous demonstrate the superior strength of the titanium screw.
Line 209: I would suggest that the mechanical resistance of the bioresorbable screw was more than 'slightly lower' than the titanium screw system.
Line 244: This requires clarification, the data would appear to suggest that 2 screw fixation offers significant advantage in both vertical and horizontal loading yet you suggest it is only advantageous for vertical forces.
The discussion would benefit from some analysis of what the required loading is for the screws. Are titanium screws overengineered for the application, may their strength and associated modulus actually result in stress shielding? Is there evidence that the strength of the bioresorbable screw isn't sufficient for application?
Author Response
Points/Contents and Responses to the Reviewer’s Suggestions
Thank you very much for your valuable comments and kind acceptance. We have incorporated all the reviewer’s comments and suggestions into our manuscript, as red letters in the revised manuscript. We would like to say thank you for the reviewer’s suggestions, which were very helpful to further, improve this manuscript.
The lists of suggestions from reviewer with answers.
Reviewer 2
The paper is a well written and well structured description of a biomechanical comparison of titanium and bioactive resorbable screw systems. A generally thorough literature review demonstrates the premise for the work, it's novelty and importance. It would be beneficial either within the literature review or discussion to mention the work of Pavlychuk et al " A comparative biomechanical evaluation of different osteosynthesis techniques used for intracapsular condylar head fractures" this paper whilst not overriding the novelty of this work does contradict the first statement within the discussion that 'This study is the first report of an in vitro biomechanical evaluation using bioresorbable screws for osteosynthesis of mandibular condylar head fractures.' The detailed comparison of the titanium and bioresorbable screw as well as the comparison of 1 and 2 screws is novel and of significant clinical value. The method was systematic and well written containing all necessary information except for description in statistics where it should be described how normality of data was determined.
Comment 1) Reviewer2: Results were clearly and concisely presented although figure 6 did not adequately show the rotation, a close up of the defect region would be beneficial.
Author response: We thank the reviewer for this helpful comment. We have enlarged the rotation of the small bone fragment in Figure 6 for clarity.
Comment 2) Reviewer2: Line 205: why does the sentence start with 'However' this sentence as in the previous demonstrate the superior strength of the titanium screw.
Author response: We thank the reviewer for this helpful comment. It is as you pointed out. We changed the conjunction to show a comparison with the HA/PLLA screw.
Comment 3) Reviewer3: • Line 209: I would suggest that the mechanical resistance of the bioresorbable screw was more than 'slightly lower' than the titanium screw system.
Author response: We thank the reviewer for this helpful comment. It is as you pointed out. We modified this sentence as follows: On the other hand, in a comparison after two-screw fixations for mandibular condylar head fractures, the bioresorbable screw model was slightly lower resistant to vertical and horizontal loads for initial displacement than that of the titanium screw model in biomechanical evaluation.
Comment 4) Reviewer2: Line 244: This requires clarification, the data would appear to suggest that 2 screw fixation offers significant advantage in both vertical and horizontal loading yet you suggest it is only advantageous for vertical forces.
Author response: We thank the reviewer for this helpful comment. It is as you pointed out. We made the following changes; This suggests that the double fixation have increased fixing force against loads in the both vertical and horizontal direction. As a noteworthy point, fixation with a single screw caused the bone fragments to turn under a high load in the horizontal load test.
Comment 5) Reviewer2: I The discussion would benefit from some analysis of what the required loading is for the screws. Are titanium screws overengineered for the application, may their strength and associated modulus actually result in stress shielding? Is there evidence that the strength of the bioresorbable screw isn't sufficient for application?
Author response: We thank the reviewer for this helpful comment. Shear strength is important as a bone cement for mandibular condylar head fractures.
In the discussion section we have added the following paragraphs.
“Shear strength was presumed to be important as osteosynthesis devices for the mandibular condylar head fractures in this study. Therefore, it is considered that the titanium screw showed higher strength than the bioresorbable screw even in the biomechanical test due to the excellent shear strength of the titanium screw itself. It is desirable to improve the shear strength in order to actively use the biomechanical screw in the future.”

Reviewer 3 Report
The manuscript titled " Biomechanical Loading Comparison Between Titanium and Bioactive Resorbable Screw Systems for Fixation of Intracapsular Condylar Head Fractures" describes mechanical resistance of titanium screws and u-HA/PLLA bioresorbable screws in single and double fixation. The article represents a major contribution to the development of new biomaterials that require specific properties such as high biological activity, steoconductivity, biocompatibility, direct binding to bone, and stable complete degradation and absorption.
In order to improve the quality of the article, the following should be considered:
Abstract
Line 18: poly-l-lactide → poly-L-lactide
Lines from 17 to 19: Delete unnecessary: ”Osteosynthesis resorbable materials made of uncalcined and unsintered hydroxyapatite (u-HA) particles, poly-l-lactide (PLLA) and u-HA/PLLA, are bioresorbable, and these materials have feasible bioactive/osteoconductive capacities.”
Keywords
Line 33: Delete text highlighting.
- Introduction
Line 44: Start the sentence with a capital letter (surgical → Surgical)
- Materials and Methods
Line 74: The authors should provide additional information on the titanium screws. Is it pure titanium or titanium alloy? It is necessary to state the titanium grade, designation of an alloy and the chemical composition.
Line 78: Term “communicate” would be better replaced with “cooperate”.
Line 99: Correct the typo (Type → type)
Line 141: Why is a 1.5 mm displacement mentioned?
Lines from 144 to 146: The authors should check the load directions in Figures 3A) and 3B). Load directions is in contrast to reference 16.
Line 153: Why is the p-value < 0.01 taken for the interpretation of the diagram, and here it is stated that a significant difference between the mean values ​​of the groups occurs if the p < 0.05. What about the results when 0.01 < p < 0.05? Are they significant or not? The article should be corrected accordingly.
- Results
Line 161: Figure 4 does not show shear strength but shear resistance.
Line 168: When the authors describe the results of the horizontal loading test they mention Figure 6 which refers to the vertical loading (according to Figure 3A). This needs to be corrected.
Lines 169 and 170: Reformulate part of the sentence "... the u-HA / PLLA screw was 1.25 times and the titanium screws were 1.32 times." because it is not clear enough. What does it refer to 1.25 times or 1.32 times? (smaller, bigger, compared to what?)
Line 171: Correct the typo (Comparison → comparison)
Lines 173 and 174: Correct the Figure number in the following sentence: "A comparison of titanium and u-HA/PLLA screws with two screws in a horizontal load test is shown in Figure 6C."
Line 177: Term “threads” would be better replaced with “screws”.
Line 182: Correct the typo (double → Double)
Line 184: Term “double screws” would be better replaced with “double fixation”
Lines from 191 to 193: Reformulate the sentence: “The measured values were the most prominent in the initial displacement amount; the u-HA/PLLA screw was 1.36 times and the titanium screw was 1.30 times that at 0.5 mm displacement." because it is not clear and does not reflect Figure 8.
Line 197: Correct the typo (Comparison → comparison)
Line 200: Term “double screws” would be better replaced with “double fixation”
- Discussion
Lines from 242 to 244: Authors should rephrase the following sentences: “Titanium screw-fixing showed a maximum load value of about 1.3 times that of their strength in vertical load and horizontal load test. The u-HA/PLLA screw showed about 1.2–1.4 times strength in vertical load and horizontal load tests.”
Lines 253 and 254: The authors should check the accuracy of the statements in the following sentence: “Fixation of double u-HA/PLLA screws showed a significantly lower resistance at the initial load compared with the two titanium screws, but this was not the case at higher loads.” because they do not reflect Figure 9.
Line 262: Term “plate” should be replaced with “screw”
References
All references must have uniquely listed pages (from the smallest to largest or vice versa).
Lines 314, 369: The year of issue should be bold.
Author Response
Points/Contents and Responses to the Reviewer’s Suggestions
Thank you very much for your valuable comments and kind acceptance. We have incorporated all the reviewer’s comments and suggestions into our manuscript, as red letters in the revised manuscript. We would like to say thank you for the reviewer’s suggestions, which were very helpful to further, improve this manuscript.
The lists of suggestions from reviewer with answers.
Reviewer 4
The manuscript titled " Biomechanical Loading Comparison Between Titanium and Bioactive Resorbable Screw Systems for Fixation of Intracapsular Condylar Head Fractures" describes mechanical resistance of titanium screws and u-HA/PLLA bioresorbable screws in single and double fixation. The article represents a major contribution to the development of new biomaterials that require specific properties such as high biological activity, steoconductivity, biocompatibility, direct binding to bone, and stable complete degradation and absorption.
In order to improve the quality of the article, the following should be considered:
Comment 1) Reviewer4: Line 18: poly-l-lactide → poly-L-lactide
Author response: We thank the reviewer for this helpful comment. we changed from poly-l-lactide to poly-L-lactide.
Comment 2) Reviewer4: Lines from 17 to 19: Delete unnecessary: ”Osteosynthesis resorbable materials made of uncalcined and unsintered hydroxyapatite (u-HA) particles, poly-l-lactide (PLLA) and u-HA/PLLA, are bioresorbable, and these materials have feasible bioactive/osteoconductive capacities.”
Author response: We thank the reviewer for this helpful comment. We removed “and u-HA/PLLA”.
Comment 3) Reviewer4: Line 33: Delete text highlighting.
Author response: We thank the reviewer for this helpful comment. We deleted text highlighting.
Comment 4) Reviewer4: Line 44: Start the sentence with a capital letter (surgical → Surgical)
Author response: We thank the reviewer for this helpful comment. We corrected to start the sentence with a capital letter.
Comment 5) Reviewer4: Line 74: The authors should provide additional information on the titanium screws. Is it pure titanium or titanium alloy? It is necessary to state the titanium grade, designation of an alloy and the chemical composition.
Author response: We thank the reviewer for this helpful comment. It is as you pointed out. We added about titanium grade.
Comment 6) Reviewer4: Line 78: Term “communicate” would be better replaced with “cooperate”.
Author response: We thank the reviewer for this helpful comment. We have replaced “communicate” with “cooperate”.
Comment 7) Reviewer4: Line 99: Correct the typo (Type → type)
Author response: We thank the reviewer for this helpful comment. We corrected it.
Comment 8) Reviewer4: Line 141: Why is a 1.5 mm displacement mentioned?
Author response: We thank the reviewer for this helpful comment. It's our mistake. It has been deleted.
Comment 8) Reviewer4: Lines from 144 to 146: The authors should check the load directions in Figures 3A) and 3B). Load directions is in contrast to reference 16.
Author response: We thank the reviewer for this helpful comment. It is as you pointed out. The figure and description were the opposite, so we fixed it.
Comment 9) Reviewer4: Line 153: Why is the p-value < 0.01 taken for the interpretation of the diagram, and here it is stated that a significant difference between the mean values of the groups occurs if the p < 0.05. What about the results when 0.01 < p < 0.05? Are they significant or not? The article should be corrected accordingly.
Author response: We thank the reviewer for this helpful comment. It is as you pointed out. The statistically significant difference is less than 0.01 in this study. The results of this study showed no difference in the items that were significantly different, even with a statistical significance of 0.05.
Comment 10) Reviewer4: Line 161: Figure 4 does not show shear strength but shear resistance.
Author response: We thank the reviewer for this helpful comment. It is as you pointed out. We have corrected for shear resistance.
Comment 11) Reviewer4: Line 168: When the authors describe the results of the horizontal loading test they mention Figure 6 which refers to the vertical loading (according to Figure 3A). This needs to be corrected.
Author response: We thank the reviewer for this helpful comment. It is as you pointed out. The vertical and horizontal diagrams in Figure 3 were reversed, so we fixed it.
Comment 12) Reviewer4: Lines 169 and 170: Reformulate part of the sentence "... the u-HA / PLLA screw was 1.25 times and the titanium screws were 1.32 times." because it is not clear enough. What does it refer to 1.25 times or 1.32 times? (smaller, bigger, compared to what?)
Author response: We thank the reviewer for this helpful comment. It is as you pointed out.
we changed it to the following sentence to make it more clear.
“u-HA/PLLA double-fixed screws was 1.25 times of single fixed screw, titanium double-fixed screws was 1.32 times of single fixed screw.”
Comment 13) Reviewer4: Line 171: Correct the typo (Comparison → comparison)
Author response: We thank the reviewer for this helpful comment. We corrected it.
Comment 14) Reviewer4: Lines 173 and 174: Correct the Figure number in the following sentence: "A comparison of titanium and u-HA/PLLA screws with two screws in a horizontal load test is shown in Figure 6C."
Author response: We thank the reviewer for this helpful comment. It's our mistake. We corrected it to Figure7.
Comment 15) Reviewer4: Line 177: Term “threads” would be better replaced with “screws”.
Author response: We thank the reviewer for this helpful comment. We replaced “threads” with “screws”.
Comment 16) Reviewer4: Line 182: Correct the typo (double → Double)
Author response: We thank the reviewer for this helpful comment. We corrected it.
Comment 17) Reviewer4: Line 184: Term “double screws” would be better replaced with “double fixation”
Author response: We thank the reviewer for this helpful comment. We replaced “double screws” with “double fixation”.
Comment 18) Reviewer4: Lines from 191 to 193: Reformulate the sentence: “The measured values were the most prominent in the initial displacement amount; the u-HA/PLLA screw was 1.36 times and the titanium screw was 1.30 times that at 0.5 mm displacement." because it is not clear and does not reflect Figure 8.
Author response: We thank the reviewer for this helpful comment. When comparing the single and double titanium screws and the u-HA/PLLA screw, there was the most prominent difference in the amount of displacement of 0.5 mm. We changed the manuscript.
Comment 19) Reviewer4: Line 197: Correct the typo (Comparison → comparison)
Author response: We thank the reviewer for this helpful comment. We corrected it.
Comment 20) Reviewer4: Line 200: Term “double screws” would be better replaced with “double fixation”
Author response: We thank the reviewer for this helpful comment. We replaced “double screws” with “double fixation”.
Comment 21) Reviewer4: Lines from 242 to 244: Authors should rephrase the following sentences: “Titanium screw-fixing showed a maximum load value of about 1.3 times that of their strength in vertical load and horizontal load test. The u-HA/PLLA screw showed about 1.2–1.4 times strength in vertical load and horizontal load tests.”
Author response: We thank the reviewer for this helpful comment. It is as you pointed out. We rephrased it as follows.
“Titanium double fixation showed a maximum load value of about 1.3 times higher than single fixation of their strength in vertical load and horizontal load test. The u-HA/PLLA double fixation showed about 1.2 to1.4 times higher than single fixation of strength in vertical load and horizontal load tests.”
Comment 22) Reviewer4: Lines 253 and 254: The authors should check the accuracy of the statements in the following sentence: “Fixation of double u-HA/PLLA screws showed a significantly lower resistance at the initial load compared with the two titanium screws, but this was not the case at higher loads.” because they do not reflect Figure 9.
Author response: We thank the reviewer for this helpful comment. It is as you pointed out. We rephrased it as follows.
“This suggests that the double fixation have increased fixing force against loads in the both vertical and horizontal direction. As a noteworthy point, fixation with a single screw caused the bone fragments to turn under a high load in the horizontal load test.”
Comment 23) Reviewer4: Line 262: Term “plate” should be replaced with “screw”
Author response: We thank the reviewer for this helpful comment. We thank the reviewer for this helpful comment. We replaced “plate” with “screw”.
Comment 24) Reviewer4: Lines 314, 369: The year of issue should be bold.
Author response: We thank the reviewer for this helpful comment. We corrected them.

Round 2
Reviewer 3 Report
Dear Authors,
Thanks for correcting the article, but some comments are still open:
- Results
Line 166: Correct the typo (3.2.1Horizontal loading test → 3.2.1 Horizontal loading test)
Lines 171 and 172 Reformulate part of the sentence "... the u-HA / PLLA doble-fixed screws were 1.25 times of single-fixed screw, whereas and the titanium doble-fixed screws were 1.32 times of single-fixed screw." because it is still not clear enough. (1.25 times / 1.32 times higher or smaller?)
Lines from 192 to 195: Reformulate the sentence: “The measured values were most prominent during the initial displacement; the u-HA/PLLA double-fixed screw was 1.36 times and the titanium double-fixed screw was 1.30 times that at 0.5-mm displacement. " because it is still not clear enough. (1.36 times / 1.30 times higher or smaller?, compared to what?)
Line 200: Correct the signature of Figure 8.
Lines 202 and 203: Please, move the signature below Figure 9.
- Discussion
Lines 245 and 246: Please, delete unnecessary in the following sentence: ” Titanium double fixation showed a maximum load value of about 1.3 times higher than single fixation of their strength in vertical load and horizontal load test.”
Lines from 246 to 248: Reformulate the sentence: “The u-HA/PLLA double fixation showed about 1.2–1.4 times higher than single fixation of strength in vertical load and horizontal load tests.” → “The u-HA/PLLA double fixation showed about 1.2–1.4 times higher resistance than single fixation in vertical load and horizontal load tests.”
References
All references must have uniquely listed number of pages (from the smallest number to largest number).
Author Response
Points/Contents and Responses to the Reviewer’s Suggestions
Thank you very much for your valuable comments and kind acceptance. We have incorporated all the reviewer’s comments and suggestions into our manuscript, as red letters in the revised manuscript. We would like to say thank you for the reviewer’s suggestions, which were very helpful to further, improve this manuscript.
The lists of suggestions from reviewer with answers.
Reviewer 4
Line 166: Correct the typo (3.2.1Horizontal loading test → 3.2.1 Horizontal loading test)
Author response: We thank the reviewer for this helpful comment. We corrected it..
Comment 2) Reviewer4: Lines 171 and 172 Reformulate part of the sentence "... the u-HA / PLLA doble-fixed screws were 1.25 times of single-fixed screw, whereas and the titanium doble-fixed screws were 1.32 times of single-fixed screw." because it is still not clear enough. (1.25 times / 1.32 times higher or smaller?)
Author response: We thank the reviewer for this helpful comment. It is as you pointed out. We have corrected the manuscript pointed out as follows.
The measured values were most prominent at 3-mm displacement: u-HA/PLLA double-fixed screws were 1.25 times higher resistance than single-fixed screw, whereas titanium double-fixed screws were 1.32 times higher resistance than single-fixed screw.
Comment 3) Reviewer4: Lines from 192 to 195: Reformulate the sentence: “The measured values were most prominent during the initial displacement; the u-HA/PLLA double-fixed screw was 1.36 times and the titanium double-fixed screw was 1.30 times that at 0.5-mm displacement. " because it is still not clear enough. (1.36 times / 1.30 times higher or smaller?, compared to what?)
Author response: We thank the reviewer for this helpful comment. It is as you pointed out. We have corrected the manuscript pointed out as follows.
The measured values were most prominent during the initial displacement; the u-HA/PLLA double-fixed screw was 1.36 times higher resistance than single-fixed screw and the titanium double-fixed screw was 1.30 times higher resistance than single-fixed screw that at 0.5-mm displacement.
Comment 4) Reviewer4: Line 200: Correct the signature of Figure 8.
Author response: We thank the reviewer for this helpful comment. We have resized Figure 8 and corrected the position of the captions in the figure.
Comment 5) Reviewer4: Lines 202 and 203: Please, move the signature below Figure 9.
Author response: We thank the reviewer for this helpful comment. We have resized Figure 9 and corrected the position of the captions in the figure.
Comment 6) Reviewer4: Lines 245 and 246: Please, delete unnecessary in the following sentence: ” Titanium double fixation showed a maximum load value of about 1.3 times higher than single fixation of their strength in vertical load and horizontal load test.”
Author response: We thank the reviewer for this helpful comment. We have replaced ” Titanium double fixation showed a maximum load value of about 1.3 times higher than single fixation of their strength in vertical load and horizontal load test.” with “Titanium double fixation showed a maximum load value of about 1.3 times higher than single fixation of their strength in vertical load and horizontal load test.”.
Comment 7) Reviewer4: Lines from 246 to 248: Reformulate the sentence: “The u-HA/PLLA double fixation showed about 1.2–1.4 times higher than single fixation of strength in vertical load and horizontal load tests.” → “The u-HA/PLLA double fixation showed about 1.2–1.4 times higher resistance than single fixation in vertical load and horizontal load tests.”
Author response: We thank the reviewer for this helpful comment. It is as you pointed out. We have revised the manuscript for that part.
Comment 8) Reviewer4: All references must have uniquely listed number of pages (from the smallest number to largest number).
Author response: We thank the reviewer for this helpful comment. It is as you pointed out. We have revised the references.
